# Remodeling of Cardiac Gap Junctional Cell–Cell Coupling

**DOI:** 10.3390/cells10092422

**Published:** 2021-09-14

**Authors:** Stefan Dhein, Aida Salameh

**Affiliations:** 1Institute for Pharmacology, University Leipzig, Härtelstr. 16, 04103 Leipzig, Germany; 2Fachdienst Gesundheit, Lindenaustr. 31, 04600 Altenburg, Germany; 3Clinic for Pediatric Cardiology, University Leipzig, Struempellstr. 39, 04289 Leipzig, Germany; aida.salameh@medizin.uni-leipzig.de

**Keywords:** gap junction, connexin, electrophysiology, cardiomyopathy, heart disease

## Abstract

The heart works as a functional syncytium, which is realized via cell-cell coupling maintained by gap junction channels. These channels connect two adjacent cells, so that action potentials can be transferred. Each cell contributes a hexameric hemichannel (=connexon), formed by protein subuntis named connexins. These hemichannels dock to each other and form the gap junction channel. This channel works as a low ohmic resistor also allowing the passage of small molecules up to 1000 Dalton. Connexins are a protein family comprising of 21 isoforms in humans. In the heart, the main isoforms are Cx43 (the 43 kDa connexin; ubiquitous), Cx40 (mostly in atrium and specific conduction system), and Cx45 (in early developmental states, in the conduction system, and between fibroblasts and cardiomyocytes). These gap junction channels are mainly located at the polar region of the cardiomyocytes and thus contribute to the anisotropic pattern of cardiac electrical conductivity. While in the beginning the cell–cell coupling was considered to be static, similar to an anatomically defined structure, we have learned in the past decades that gap junctions are also subject to cardiac remodeling processes in cardiac disease such as atrial fibrillation, myocardial infarction, or cardiomyopathy. The underlying remodeling processes include the modulation of connexin expression by e.g., angiotensin, endothelin, or catecholamines, as well as the modulation of the localization of the gap junctions e.g., by the direction and strength of local mechanical forces. A reduction in connexin expression can result in a reduced conduction velocity. The alteration of gap junction localization has been shown to result in altered pathways of conduction and altered anisotropy. In particular, it can produce or contribute to non-uniformity of anisotropy, and thereby can pre-form an arrhythmogenic substrate. Interestingly, these remodeling processes seem to be susceptible to certain pharmacological treatment.

## 1. Introduction and General Considerations

Electrical activation is a key phenomenon in cardiac physiology. The heart beats in a regular rhythm determined by the sinus node, transferred to the atrium, the AV-node, the bundle of His, the Tawara branches and finally via the Purkinje fibers to the working ventricular myocardium. While in the Purkinje fibers the action potential propagation velocity is about 0.8–1.0 m/s, in the working myocardium the conduction velocity (CV) along the fibers is about 0.4 m/s and transverse to the fiber axis is about 0.2 m/s. This difference in CV is due to the architecture and cell form of the cardiomyocytes and the distribution of the gap junctions which are mainly found at the cellular poles. The difference in CV between longitudinal and transverse conduction-named anisotropy- can exist as uniform or non-uniform anisotropy, the latter being typical for aged hearts or hearts with scars or with enhanced content of fibrous tissue [1,2] (Spach & Dolber, 1990; Dhein & Hammerrath, 2001). 

Typically, an action potential is transferred from one cell to another via the gap junction channels, which are mainly localized at the cell poles in the plicate and interplicate region (Dolber et al., 1992) [3]. Although in the classical experiments by Weingart and Maurer (1988) [4], it was shown that the successful cell-to-cell conduction of an action potential requires at least > 13 gap junction channels between the cells, there is also evidence that under certain pathological conditions with reduced coupling ephaptic coupling may occur (Veeraraghavan et al., 2015; George et al., 2019; Gourdie, 2019), which might represent a kind of back-safe mechanism [5,6,7]. Recent studies have shown that ephaptic coupling requires an electrical field sufficient for the activation of clusters of sodium channels, and that the clustering of sodium channels potentiates the ephaptic interaction, which is even higher if the gap junction coupling is reduced [8] (Hichri et al. 2018). This may contribute to the formation of areas with slow conduction as a typical arrhythmogenic factor [9] (Carmeliet, 2019). The number of gap junction channels is inversely correlated to the conduction velocity [10] (Seidel et al., 2010). However, under normal normoxic conditions, some experiments have shown that a 90% decrease in the number of gap junctions is necessary to reduce conduction velocity by 25% [11] (Jongsma and Wilders, 2000). This oversimplifies the situation a bit, since the effectivity of conduction not only depends on the pure number of gap junction channels but also on their single channel conductance, on the relationship between this number of channels, and the sizes of the activated cell and of the cell /area which is to be activated. 

If a cell is electrically activated, it can be considered as a “current source” while the still un-activated neighboring cell is a “current sink”. However, if the current sink is too large, the current source may be too small to activate this area (this problem is known in the literature as the “current source/sink problem” [12,13,14,15,16,17,18,19,20,21] Rohr et al., 1997; Rohr, 2004, 2012; Lee and Pogwizd, 2006). The voltage difference between these two cells resembles the driving force for this current, flowing via the gap junction channels and, to some extent, via the extracellular space. The quantity of current transferred is determined by the geometric properties of the tissue. This means a small source can activate only a very limited number of neighboring cells. The sinus node is a good example of this phenomenon: the sinus node is a tiny current source surrounded by a large sink (atrium). Conduction is maintained by a limited expression of the gap junctions localized in interdigitating finger-like zones extending from the sinus node to the atrium (Joyner & van Capelle, 1986; Boyett et al., 2006) [16,17]. In a more general or mathematical form this can also be described as the ratio of charge produced/charge consumed, which has been defined as the safety factor (SF) of propagation: SF = Q_c_ + Q_out_/Q_in_(1)

Qc is the membrane capacitive charge; Q_out_ is the outward charge transfer; Q_in_ is the inward charge transfer, with both resulting from the ionic currents Shaw & Rudy, 1997a,b [18,19]. As long as SF is > 1, propagation will be successful.

Taken together, this means that geometrical properties are very important for conduction. Problematic situations causing source-sink problems are e.g., a high curvature of the propagation wave front, the endings of the Purkinje fibers, the areas in which the activation front passes an isthmus, encircles obstacles, or the border between ischemic (non-excitable tissue; current sink) and non-ischemic tissue (excitable tissue; current source), and also the border between cardiac tissue and fibrous strands. This all has to be considered together with the cell size of cardiomyocytes, the cell form, and the distribution and density of the gap junction channels with respect to the cell axis. Computer simulations have revealed that that an increased cell diameter (as in hypertrophied cells) may accelerate the longitudinal propagation velocity θ_L_, but only if the gap junction conductance (g_GJ_) is enhanced proportionally to the cellular membrane surface A_m_. If g_GJ_ remains constant, an increase in the cell diameter leads to a reduced θ_L_ (Seidel et al., 2010; Dhein et al., 2014) [10,20].

Research in the last few years has revealed that the cells actively adapt their gap junctions to the specific situation. The expression and the subcellular localization of the gap junction channels, and their constituting proteins are highly dynamic and change in the course of cardiac disease.

Gap junction channels are formed by two hexameric hemichannels contributed by each of the two coupling cells. The hemichannels consist of six proteins called connexins. Connexins exist in 21 isoforms of various molecular weight (Cx stands for “connexin”, the number gives the approximate molecular weight in kDa), which mainly depends on the length of the C-terminus. Cardiomyocytes express the isoform Cx43, Cx40, and Cx45. While Cx43 is ubiquitous in the heart, Cx40 is mainly restricted to the atria and the conduction system, and Cx45 is expressed mainly in earlier developmental stages. Besides these, in the vasculature, Cx37 is found. In the sinoatrial node Cx43 is not expressed, but Cx30.2 and Cx45 are expressed. However, in the interdigitating finger-like zones, Cx43 and Cx45 are present (Boyett et al., 2006) [17]. In the atrioventricular node and the bundle of His, Cx45 is most important for conduction, modulated by Cx30.2 and Cx40 (Schrickel et al., 2009) [21].

## 2. Regulation of the Acute Opening and Closure of Gap Junction Channels

Gap junction channels can switch between various states: a closed state, an open state with several subconductance states (e.g.,in the case of Cx43 three: 30, 61, 89 pS [22] (Kwak et al., Mol Biol Cell 1995), a closed state (0 pS), and a residual state (3 pS; [23] Valiunas et al., 1997) (See also [24] Bukauskas and Verselis, 2004). The preferential subconductance state seems to depend on the phosphorylation / de-phosphorylation status of the connexin at certain residues, in particular within the C-terminal. Besides the single channel conductance, the mean open time and open probability [25] (Brink et al., 1996) and the transition time (from open to closed state and vice versa) can also be regulated [26] (Bukauskas and Peracchia, 1997). Generally, the macroscopic conductance (g_j_) is affected by the number of channels (N), their unitary conductance (γ_j_), and their open probability (P_o_) as given by the equation: g_j_ = N × γ_j_ × P_o_(2)

The important regulators of gap junction conductance are pH, intracellular calcium, and transjunctional voltage. Thus, intracellular acidification results in the closure of the gap junction channels. Among others, two main processes are responsible: upon a drop in the pH the C-terminal changes its conformation, thereby closing the channel pore in a “ball-and-chain-like mechanism” [27] (Ek-Vitorin et al., 2016), and secondly, H^+^ protons can displace Ca^2+^ from its binding sites leading indirectly to gap junction closure via increased [Ca^2+^] [28] (Peracchia, 2004). On the other hand, the N-terminal is involved in voltage sensing and connexin channel gating [29] (Verselis et al., 1994). The closure of the Cx43 gap junction channels by elevated [Ca^2+^]_i_ is thought to be mediated via Ca^2+^-calmodulin signaling and a calmodulin interactive site at the C-terminal [28] (Peracchia, 2004). Phosphorylation of the C-terminal can change the macroscopic gap junction channel conductance [30] (Moreno and Lau, 2007). PKC has been reported to (a) reduce the unitary event activity from 100 pS to 50–60 pS events [31] (Lampe et al., 2000) and (b) to increase the open probability [32,33] (Kwak et al., 1995; Kwak and Jongsma, 1996), so that the resulting net effect is a mixture, which may depend on the PKC isoforms involved. For more details on this issue see the review by Leybaert et al. [34] (Leybaert et al., 2017).

A number of stimuli can regulate gap junction macroscopic conductance (see Table 1). Many of these stimuli play an important role in the closure of the gap junctions during acute cardiac ischemia, such as ATP loss, decrease in pH, acylcarnitines, etc. In ischemia, the acute closure of the gap junctions prevents the cardiomyocytes from further loss of ATP and further oxygen consumption. Cells become electrically silent and isolated which —on the other hand—changes the geometry of activation propagation and may produce arrhythmia (Dhein, 2006; Jozwiak & Dhein, 2008; De Groot & Coronel, 2004) [35,36,37]. Since the focus of the present review is the remodeling processes in cardiac disease, we refer for a more specific in depth description of the acute processes to the literature (e.g., Leybaert et al., 2017) [34].

## 3. Regulation of the Expression of Gap Junction Proteins

The constituents of the gap junction channels are connexins. These are proteins which are synthetized in the sarcoplasmic reticulum, then transported in small vesicles to the Golgi apparatus. A connexin protein has an intracellular C- and N-terminus, four transmembrane domains, and two extracellular loops, the latter being stabilized by intramolecular disulfide bonds. In the trans-Golgi apparatus, connexins are assembled to hexameric subunits, the so called connexons or hemichannels. These are subsequently transported to the plasma membrane and are integrated in the lipid bilayer in domains rich in N-cadherin and zonula occludens protein ZO-1 (Li et al., 2016; Musil & Goodenough, 1993, 1995; Falk, 2000; Giepmans, 2004) [38,39,40,41,42].

To understand the remodeling processes with regard to gap junctional coupling, an important aspect is the regulation of cardiac connexin expression, i.e., the expression of Cx40, Cx43, and Cx45.

There are a number of signaling pathways which have been identified to be involved in the regulation of connexin expression. As a general rule it can be stated that pathways which are linked to the hypertrophy of cardiomyocytes are also typically associated with enhanced connexin expression, mainly concerning Cx43. The pathways and the effects on connexin expression are given in Table 2.

Finally, gap junction channels are degraded with a short half-life time of ca. 90 min (Cx43) via phosphorylation, ubiquitinoylation, and proteasomal degradation. In some cases, lysosomal degradation has also been described (Berthoud et al., 2004; Salameh, 2006) [43,44]. Thus, there is a dynamic equilibrium between the integration and degradation of gap junction channels. The continuous turnover of the gap junction channels allows the cell to incorporate newly synthetized channels at other locations and to adapt not only the extent of coupling but also to the localization.

## 4. Regulation of the Localization of Gap Junction Channels

The next aspect of the remodeling processes concerning gap junctional coupling is the regulation of the subcellular distribution and the localization of cardiac connexins. Cardiac tissue is composed of cardiomyocytes in a bricklike arrangement with overlapping cell endings. Each cell is characterized by a long axis and a much smaller diameter. This is one aspect of the anisotropy since this means a lower longitudinal resistance in comparison to the transversal resistance. Gap junction channels are located predominantly at the cell poles in the plicate and interplicate regions (Dolber et al., 1992) [3], so that an action potential will propagate faster in the longitudinal than in the transverse direction. The fibers of the specific conduction system do not communicate with the surrounding cardiac tissue in the septal area. The conduction system communicates mainly via the gap junction channels consisting of Cx40, while the working myocardium communicates via the Cx43 channels. Only at the endings of the Purkinje fibers (about the last third) does the coupling between the conduction system and the working myocardium occur (Olejnickova et al., 2021) [45].

In the past few years, research has focused on the question of how the localization of gap junction channels may be regulated. Using an EGFP-tagged Cx43 (EGFP = enhanced green fluorescent protein) Bukauskas and colleagues (2000) [46] showed that channels form plaques by clustering and that in plaques of about 0.5 µm in diameter, about 2000 channels can be found, of which 10 to 20% are functional. In smaller plaques this fraction is lower and tends to be zero. These authors assumed that channels may cycle between an active and inactive state, which may be regulated by phosphorylation processes. Critically, one needs to state that it cannot be totally excluded that the EGFP-tag itself may affect the life cycle of connexin, the availability of phosphorylation sites or the geometrical features regarding the clustering in plaques. However, these observations suggest that very small plaques of connexins might be too small to contain a sufficient number of function channels to allow action potential transfer.

Gap junction channels are clustered in the intercalated disk (ID) in close neighborhood to the sodium channels [47] (Moise et al., 2021). This would allow the elicitation of an action potential at the earliest moment when the current flowing through the gap junctions depolarizes the membrane in the ID area. However, the ID is not a simple homogenous structure but has a complicated pattern with nanodomains which can enhance or attenuate gap junction coupling. Moreover, this explains why a successful transfer of action potentials is not simply granted by a certain number of gap junctions, but the interaction of ID nanodomains, gap junction clustering, and interaction with neighbored sodium channels [21] (Moise et al., 2021).

Interestingly, it has been shown that a cyclic mechanical stretch can activate cardiomyocyte hypertrophy gene programs (De Jong et al., 2013) [48] and can induce the accentuation of gap junction channels (Cx43) and N-cadherin at the cell poles [49]. This process was reversible within 24 h and could be prevented by MEK1/2 inhibition. A cyclic mechanical stretch also enhanced Cx43 expression and up-regulated the phosphorylated forms of ERK1/2, glycogen synthase kinase 3beta, and AKT (Salameh et al., 2010) [49]. Further investigations revealed that the stretch signal sensing is mediated via focal adhesion kinase (FAK) and leads to intracellular re-organization and orientation involving the microtubules and the Golgi apparatus (Dhein et al., 2014) [50]. A cyclic mechanical stretch led to an accentuation of the Golgi apparatus together with the microtubule organizing center (MTOC) at one site of the nucleus facing a cell pole with the nucleus facing the opposite cell pole. Kinesin, the plus motor protein, was found mainly at the endings of the microtubule near the cell poles and the cell periphery. Dynein, the minus motor protein, was found at the origin of the microtubule near to the Golgi apparatus. Moreover, the cell axis orientation depended on the stretch axis (Dhein et al., 2014) [50]. The effects of a cyclic mechanical stretch can be modulated by angiotensin II (Salameh et al., 2012) [51] and by alpha- and beta-adrenoceptor stimulation (Salameh et al., 2010a) [52]. Figure 1 summarizes the effects of cyclic mechanical stretch on the localization of cardiac gap junctions.

With regard to the cardiac remodeling of gap junctions, this may be an important factor, since in most cardiac diseases altered mechanical forces, stretch, and the altered direction of stretch as well as mediators stimulating angiotensin-receptors and adrenoceptors are involved.

## 5. Gap Junction Remodeling in Atrial Fibrillation (AF)

AF is a common arrhythmia in older patients. The typical origin of this arrhythmia is the left atrium near the confluence of the pulmonary veins. Anatomically, the heart is fixed in the mediastinum at the pulmonary veins and the caval veins, so that high mechanical forces result at the insertion of the veins from the regular beating of the heart. It is not known whether this chronic cyclic stretch is involved in the initiation of AF, but one might imagine that this type of stretch could activate stretch activated ion channels (Ninio & Saint, 2008) [53], which then may provoke action potentials in this area (Hocini et al., 2002; Po et al., 2005; Teh et al., 2011) [54,55,56]. Interestingly, this is the area where a static region (pulmonary vein) is directly neighbored to a highly dynamic regularly beating region (atrium). In this area, sleeves protruding from the atrium into the veins forming circular structures coupled by Cx43 and to a lower extent by Cx40 have been observed and were assumed to serve as a structural basis for a rotor wave (Verheule et al., 2002) [57]. For a complete overview of the factors involved in initiation of AF see (Franz et al., 2012; Harada & Nattel, 2021; Nattel et al., 2020) [58,59,60]. After AF is initiated, arrhythmic and irregular wavelets travelling over the atrium occur. The high frequency of action potentials during the fibrillation episodes causes increasing intracellular calcium concentrations resulting in Ca^2+^ handling abnormalities. This activates a plethora of pathways such as CamKII, PKC, MAPK, PKA, JNK, NFAT, and GATA4, which leads to the altered expression of various ion channels, ion transporters/exchangers, and intracellular proteins, finally resulting in shortened action potentials which facilitate atrial fibrillation (for review see Nattel et al., 2020) [60]. Besides this, autocrine and paracrine pathways involving IL-1β, IL-6, and CamKII are activated leading to fibrosis, hypertrophy, and apoptosis. The serine protease cathepsin A also contributes to the remodeling process at least in diabetes mellitus (Linz et al., 2016; Hohl et al., 2019) [61,62]. AF leads to an increased deposition of collagen I, and if associated with mitral valve disease, leads to an increased deposition of collagen III (Boldt et al., 2004) [63]. Moreover, the strands of non-excitable fibrous tissue enhance the irregularity of anisotropy and facilitate the induction of atrial fibrillation (Angel et al., 2015) [64]. The longitudinal conduction velocity in the atrium has been measured and a value of 0.55 m/s was obtained, and the value of the transverse conduction velocity was 0.25 m/s (Krul et al., 2015) [65].

Taken together, AF is typically self-stabilizing as summarized in the famous sentence “atrial fibrillation begets atrial fibrillation” (Wijffels et al., 1995) [66]. An important point in this process is the remodeling of intercellular coupling.

Thus, it was shown that in patients with chronic AF, Cx40 was enhanced, while Cx43 remained unchanged. However, these changes were accompanied with a lateralization of both the Cx43 and Cx40 proteins (Polontchouk et al., 2001) [67]. Depending on the type of AF, others also found an enhanced Cx43 expression if AF was caused by mitral valve disease (Wetzel et al., 2005) [68]. Although the level of expression of connexins seems to depend on the type of AF (lone AF, AF with mitral valve disease, AF with coronary heart disease etc.), the dislocation of the gap junctions from the cell poles to the lateral sides, first described by the authors of (Polontchouk et al., 2001) [67], has subsequently been seen by others (Kostin et al., 2002; Li et al., 2004) and seems to be a general feature of gap junction remodeling in AF [69,70]. Besides this, the expression level becomes regionally different in AF (Dupont et al., 2001) [71]. In patients, there was no significant gender effect on the changes in AF-induced Cx43 and Cx40 (Pfannmüller et al., 2013) [72]. However, in some animal models, changes in connexin expression in induced AF were lacking, e.g., in the goat model (Neuberger et al., 2005) [73].

If these channels are functional, one should expect that the transverse conduction velocity should be enhanced in AF and anisotropy might be reduced. Thus, in a rat atria model, a reduction in anisotropy and an increase in the transverse velocity could be induced within 24 h of fibrillation accompanied by connexin lateralization (Polontchouk et al., 2001) [67]. In human atrial tissue, mapping experiments also revealed an increase in transverse conduction velocity and a decrease in anisotropy in AF with a more irregular conduction pattern (Dhein et al., 2011) [74]. Irregularity in conduction also seems to be common in AF (Colman et al., 2014) [75].

It remains unclear what the leading factor is that provokes these changes in gap junction location. However, AF also leads to intracellular remodeling of the Golgi-microtubular apparatus. Thus, in patients with chronic AF (duration > 1 year) fragmentation of the Golgi apparatus together with a reduced fragment size was reported. Most of the Golgi fragments were observed lateral to the nucleus. In contrast, in sinus rhythm (SR) the Golgi apparatus is also located near the nucleus but is orientated towards the cell pole following the longitudinal axis of the cell. In addition, in these AF patients the number of tubulin strands longer than 10 µm was reduced, which was associated with an increased membrane association of cdk5, but not with the activation of stathmin (Jungk et al., 2019) [76]. Stathmin can stop the growth of the microtubuli at their endings, which—in neuros—can induce Golgi fragmentation (Bellouze et al., 2016) [77]. Cdk5 is known to induce Golgi fragmentation by the phosphorylation of GM130 (Sun et al., 2008) [78]. Thus, in AF, the fragmentation of the Golgi seems to be mediated via cdk5 (Jungk et al., 2019) [76]. These observations indicate that AF leads to a very complex intracellular remodeling which may form the basis for the altered orientation of gap junction localization.

Theoretically, one may imagine that enhanced stretch in AF (e.g., caused by mitral valve disease, or by the irregular movements and contractions) may deflect integrins bound to collagen fibers, which subsequently activates FAK. This can result in the activation of ERK1/2 (Salameh et al., 2010) [52] and may lead to changes in connexin expression and via the re-organization of Golgi apparatus and microtubules (Dhein et al., 2014) [50] to alterations in the intracellular protein transport system and thus in the final localization of the gap junction channels.

Angiotensin II has been considered as a possible factor involved in atrial remodeling in AF. Chronic atrial fibrillation is associated with an up-regulation of AT(1) in the left, but not in right atrium (Boldt et al., 2003) [79]. ACE-inhibitor therapy reduced AF-induced fibrosis (Boldt et al., 2006) [80]. Further investigations in cultured cardiomyocytes revealed that the stretch-induced self-organization of cells with cell elongation and polarization of Cx43 could be modulated by angiotensin II: angiotensin II enhanced Cx43 expression via AT(1)-receptors but reduced Cx43 polarization via AT(2)-receptors (Salameh et al., 2012) [51]. Regarding the underlying mechanism, angiotensin II seems to activate CTGF via the activation of Rac1 and nicotinamide adenine dinucleotide phosphate oxidase. This results in an increase in Cx43, N-cadherin, and interstitial fibrosis (Adam et al., 2010) [81].

Beta-adrenoceptor stimulation has been shown to result in an up-regulation of Cx43 expression (see Table 2, Salameh et al., 2006, 2009) [82,83]. The reduced expression and lateral distribution of Cx43 in cardiomyocytes after β-adrenoceptor stimulation was also shown by others which interestingly was associated with the inverse effects in cardiac fibroblasts (Zhang et al., 2020) [84]. Previously, it was shown that cardiac fibroblasts inhibit β-adrenoceptor-dependent Cx43 signaling in cardiomyocytes involving angiotensin II (Salameh et al., 2013) [85].

With regard to AF, in the left atria from 160 patients with either chronic AF or sinus rhythm, enhanced Cx43 and Cx40 expression was observed together with the lateralization of both connexins. This AF-induced increase in the lateral/polar expression of Cx43, but not of Cx40, was significantly diminished in patients receiving metoprolol. In good accordance with this data, the mapping of atria from AF patients in comparison to those from patients with sinus rhythm revealed that the functional correlate transverse conduction velocity was significantly enhanced in AF patients and that this change was also significantly reduced in those receiving metoprolol, indicating a role for ß-adrenoceptors in this process (Dhein et al., 2011) [74].

These mechanisms are illustrated in Figure 2.

## 6. Gap Junction Remodeling in Ischemic Heart Disease

Acidosis, ATP-depletion, Ca^2+^ or Na^+^ overload, as well as acylcarnitine and lysophosphoglycerine lead to acute gap junction uncoupling. These are all factors enhanced in ischemia reperfusion injury. Acutely, this results in conduction block, electrical inactivity, and the isolation of the ischemic area. This mechanism can protect from further loss of ATP in this area, but on the other hand, it provokes changes in the propagation of action potentials, which can result in arrhythmia. Typically, the ischemia-induced closure of the gap junction occurs within 15–20 min and the resulting ventricular arrhythmia occurs about 20–30 min after coronary occlusion (Jozwiak & Dhein, 2008; Hagen et al., 2009) [36,86], and is associated with Cx43 de-phosphorylation (Jeyamaram et al., 2003) [87]. Cx43 dephosphorylation at S282 was identified as a factor leading to arrhythmias and a factor involved in cardiomyocyte death (Xue et al., 2019) [88]. Recently, EHD1 (Eps15; endocytic adaptor epidermal growth factor receptor substrate 15) has been shown to enhance the ischemia-induced lateralization of Cx43 in cardiomyocytes (Martins-Marques et al., 2020) [89].

After the survival of the acute phase of ischemia, a complex remodeling process starts: the infarcted myocardium heals through the replacement of the damaged cardiomyocytes by fibrous tissue. This process includes an inflammatory response, the proliferation of fibroblasts, the formation of an extracellular matrix (ECM), the processing of ECM by metalloproteinases and cathepsins, as well as hypertrophic changes of the surviving cardiomyocytes (Frangogiannis & Kovacic, 2020) [90]. Macrophages and fibroblasts play a central role in the orchestration of this process (Peet et al., 2020) [91]. The remodeling also affects intercellular coupling not only between the cardiomyocytes, but also between cardiomyocytes and fibroblasts (Camelliti et al., 2006) [92]. Metalloproteinases are involved in the cleavage of Cx43 and inflammatory cytokines such as IL-1ß contribute to the regulation of Cx43 expression (Mouton et al., 2018; De Jesus et al., 2017) [93,94]. In particular, in the border zone Cx43 is no longer restricted to the cell poles but exhibits a dispersed pattern. In the few last years, it has become obvious that cardiomyocytes can also couple to fibroblasts and this might be of particular importance in the chronic phase of myocardial infarction (Camelliti et al., 2004) [95]. Moreover, it has been demonstrated that the heterocellular coupling between cardiomyocytes and fibroblasts is functional (Quinn et al., 2016) [96]. The contacts between cardiomyocytes and myofibroblasts are not static but change with time due to the dynamics of the myofibroblast lamellipodia (Schultz et al., 2019) [97]. In the chronic phase of infarction, typical changes are loss of the polarized organization of Cx43 and a reduction in Cx43 expression (Severs et al., 2006) [98]. Over a distance of up to 300 µm, a successful but delayed conduction between cardiomyocytes separated by fibroblasts can be observed (Rohr, 2004) [8]. As a consequence of the scar formation after myocardial infarction, the electrical properties of the tissues become more inhomogeneous or non-uniform. Thus, excitable cardiomyocytes are closely adjacent to non-excitable fibrous tissue. This architecture changes the current source to sink ratio regionally. Since there is less current flowing to the adjacent fibrous tissue as to the adjacent myocytes, an action potential which propagates towards a non-excitable zone will exhibit longer duration in those cardiomyocytes close to the non-excitable area than in those surrounded only by excitable tissue, so that there is a dispersion of the action potential duration (Gottwald et al., 1998; Müller & Dhein, 1993) [99,100]. Moreover, the spread of excitation will be altered (Kucera et al., 2017) [101]: the activation wavefront can become irregular or zig-zag-formed, and its propagation will be delayed at the fibrotic strands, which is mirrored in fractionated extracellular electrograms (Spach et al., 1982) [102]. However, a fibrous strand does not necessarily mean failure of propagation. If only a few fibroblasts are connected to a larger number of cardiomyocytes, delay of propagation will probably occur. However, if only a small surviving area of cardiomyocytes is connected downstream of the activation direction to a large excitable area, the current loss to that area (current sink) may be large enough that a conduction delay or even failure occurs (Kucera et al., 2017) [101]. In such a situation, it will be interesting to investigate the substructure of the intercalated disks, the nanodomains, the Cx43 clustering, and the fraction of functional channels, as well as the question of how many sodium channels are present around these plaques [47]. With regard to the results of the Kucera group [8], one could speculate that in such a tissue, ephaptic coupling may occur and contribute to slow conduction [9]. Mismatches of the source-to-sink ratio are associated with curved wavefronts of activation. Furthermore, non-cardiomyocytes and cardiomyocytes can interact electrotonically, and fibroblasts can establish gap junctional coupling to cardiomyocytes leading to a delayed conduction and ectopic activity (Kucera et al., 2017; Miragoli et al., 2006, 2007) [101,103,104]. Cx43 plays an important role in these heterocellular gap junctions (Schultz et al., 2019) [97]. These gap junction channels can be homotypic Cx43 and Cx45, and heterotypic Cx43/Cx45, and Cx45/Cx43. However, it has been shown that these different channel isoforms do not differ functionally with regard to the conduction velocity (Brown et al., 2016) [105]. Taken together, all these factors form an arrhythmogenic substrate and can finally lead to re-entrant arrhythmia. The higher grade in non-uniformity and inhomogeneity of gap junctional coupling, the irregularity of tissue excitability (by non-excitable tissue), and local source-to-sink mismatches are among the important characteristics of this post-infarction tissue.

## 7. Gap Junction Remodeling in Cardiomyopathy

Cardiomyopathy is a serious cardiac disease which can lead to heart failure and to ventricular arrhythmia. Typical causes of cardiomyopathy are (a) ischemia, infarction, and coronary heart disease, (b) toxic effects (e.g., by certain cytostatic drugs such as doxorubicin), (c) inflammatory and immunological processes including valve disease, (d) chronic arterial hypertension, or (e) hypertrophic obstructive cardiomyopathy (HOCM).

The pump efficacy in cardiomyopathy is reduced, which is compensated for a long time by hypertrophic processes concerning the cardiomyocytes. However, along with the changes of the cell size, changes in Ca^2+^ handling, in ion channels, ion pumps, and in cytoskeleton have also been observed. Regarding the latter, changes in connexin expression have also been identified. Thus, an upregulation of Cx43 was described in compensated hypertrophy, while in decompensated hypertrophy diminished and heterogeneous Cx43 distribution was found (Severs et al., 2006; Kostin et al., 2004) [98,106]. In a subsequent study it could be demonstrated that the reduction in Cx43 was associated with a decrease in ZO-1 (Kostin, 2007) [107]. In the early stage of heart failure Cx43 was localized to lateral sides of the cardiomyocytes (Kostin et al., 2004) [106]. In end-stage failing human hearts, the reduced Cx43 expression was associated with diminished Cx43-N-cadherin co-localization and with reduced transmural conduction velocity indicating functional relevance (Glukhov et al., 2012) [108]. Moreover, these authors observed wave breaks due to fibrotic strands in the failing hearts. However, in contrast to the findings in early heart failure mentioned above, in these 10 end-stage failing hearts no lateralization of Cx43 was observed (Kostin et al., 2004; Glukhov et al., 2012) [106,108]. However, others have also described lateralization in non-ischemic dilated cardiomyopathy in a rapid pacing model (Akar et al., 2004) [109]. In non-failing hearts of aged rabbits, the lateralization of Cx43 together with fractionated extracellular electrograms and irregular propagation of activation with reduced anisotropy and enhanced transverse conduction velocity was also observed (Dhein & Hammerrath, 2001) [2]. Thus, the type of cardiomyopathy and the stage of the disease may modulate the gap junction remodeling process.

Further investigations revealed that β-adrenoceptor stimulation up-regulated cardiac Cx43 expression via a protein kinase A and MAPK-regulated pathway, possibly involving AP1 and CREB. In patients with DCM, Cx43 expression was significantly lower, while in patients with HOCM, Cx43 content was significantly higher, as compared to patients without any cardiomyopathy. Cellular Cx43 distribution was also altered in cardiomyopathy with more Cx43 being localized at the lateral border of the cardiomyocytes (Salameh, Krautblatter et al., 2009) [83].

Taken together, these observations suggest that in heart failure, reduction in Cx43 expression seems to be a common phenomenon and in early stages, the lateralization of Cx43 occurs while in late stages, fibrosis seems to be more important. A general adaption phenomenon, however, is the hypertrophy of the cardiomyocytes. Hypertrophy alters not only the cell size but also its biophysical properties by a change in the membrane surface A_M_ and electrical capacitance in relation to the gap junction conductance (g_GJ_). Biophysical considerations using computer simulations revealed that cell size (length and width of a cell) affects the sensitivity to uncoupling. Gradual uncoupling leads to faster reduction in longitudinal conduction velocity in wide cells (Seidel et al., 2010) [10]. Due to the overlap of cells, an increase in lateral gap junction conductance stabilizes the cells against uncoupling and can balance out inhomogeneities of activation. Thus, the lateralization of gap junctions in cardiac hypertrophy may—at least in parts—represent a compensatory mechanism (Seidel et al., 2010) [10]. On the other hand, together with irregularly distributed collagen strands, local situations may occur, in which increased cell size, lateral gap junction coupling, increased cell capacitance (which needs to be loaded by the activated cells to enable propagation of the impulse), and an obstacle such as fibrous tissue produce slowed conduction and increased action potential duration differences (i.e., enhanced dispersion), and may thereby allow a circular movement of the activation wavefront, finally leading to ventricular fibrillation. Similar to the interactions between the cytoskeleton and Cx43 localization described above (see Figure 1), Macquart and colleagues (2019) [110] also noted that microtubules regulate the Cx43 lateralization in certain forms of cardiomyopathy (see also: Trembley et al., 2018) [111].

In addition, some of the lateral connexins may not represent complete functional gap junctions but Cx43 hemichannels. These have been shown to be coupled to Ca^2+^ release and Ca^2+^ dynamics, so that in failing hearts hemichannel opening can contribute to delayed afterdepolarizations, triggered action potentials, and electrical instability (De Smet et al., 2021) [112]. Further investigations have demonstrated that the activation of ryanodine receptors (Ry2R) can open Cx43 hemichannels (Lissoni et al., 2021) [113].

In a mouse model of right ventricular pressure overload, it was recently shown that cardiac macrophages are involved in the maintenance of cardiac gap junctional communication and impulse conduction by the release of amphiregulin which controls connexin 43 phosphorylation and translocation in cardiomyocytes (Sugita et al., 2021) [114]. The lateralization of gap junctions also occurs in the hearts of Duchenne muscular dystrophy. This has been linked to Cx43-S325/S328/S330 serine hypophosphorylation [115] (Himelman et al., 2020).

## 8. Concluding Remarks

Gap junction remodeling in cardiac disease involves changes in connexin expression, resulting in altered gap junction density and functional coupling, as well as changes in gap junction subcellular localization. The cardiac tissue is a complex network of excitable (cardiomyocytes) and non-excitable cells (fibroblasts etc.), and acellular, electrical passive tissue (fibrous strands). The functional architecture is defined by cardiomyocyte cell size (which may change with cardiac disease), by the number, function, and localization of electrical contacts (gap junctions), the non-excitable areas and their electrical resistance, the ratio between gGJ, longitudinal resistance, transverse resistance, cell capacity, and capacity of the cell, which is to be activated next. Since source-sink problems also are important for propagation, the direction of conduction may alter the effects of a given tissue architecture: a small, activated area meeting a large (non-activated) sink area may result in conduction failure, while propagation in the opposite direction may be possible. This will be critically modulated by other factors such as GJ distribution, fibrous tissue, cell size, etc.

The remodeling of this architecture in cardiac disease is not a simple up or down mechanism regarding conduction but a very complex interplay. Nevertheless, atrial fibrillation, cardiac ischemia, chronic infraction, altered stretch, and cardiomyopathy all lead to changes of this architecture, which can form an arrhythmogenic substrate. Thus, a reduction in coupling and the changes in tissue biophysics as described above will lead to a situation with an enhanced local dispersion of action potential duration or of refractoriness together with a slowing of conduction, both favoring re-entrant arrhythmia. The action potentials of a several isolated cardiomyocytes normally differ regarding their duration. If these cells are coupled via gap junctions these differences are diminished. Differences in action potential duration result in a voltage gradient between the cells, which will cause a gap junction current between these cells. This will lead to the assimilation of the potential durations. Thus, local dispersion or inhomogeneity are kept at a minimum by gap junction coupling. Some of the changes related to hypertrophy can be interpreted as a compensatory mechanism to adapt cellular coupling to the changes in cell size, resulting however in the formation of an arrhythmogenic substrate.

## Figures and Tables

**Figure 1 cells-10-02422-f001:**
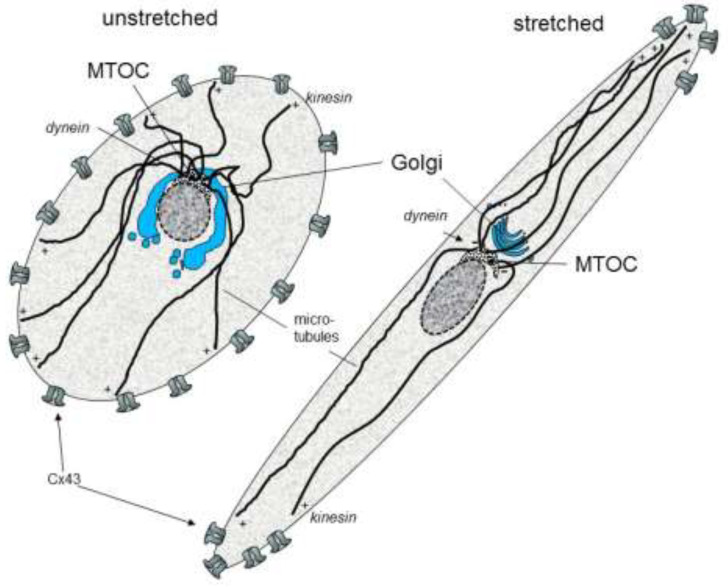
Schematic effects of cyclic stretch on the organization of cytoskeleton and gap junctions in cardiomyocytes (MTOC: microtubule organizing center; Cx: connexin).

**Figure 2 cells-10-02422-f002:**
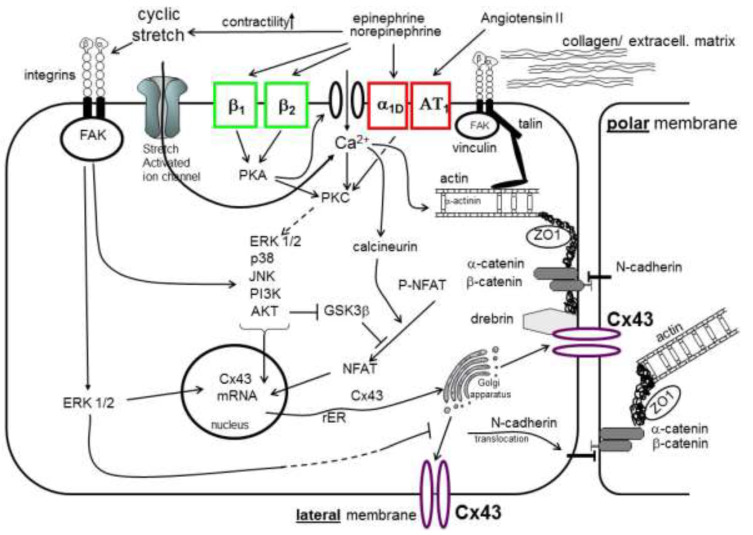
Gap junction remodeling processes in cardiomyocytes (abbr.: see list).

**Table 1 cells-10-02422-t001:** Stimuli affecting acute macroscopic gap junction conductance (C = closure; O = opening or increase in conductance; other abbr.: see list).

Stimulus	Opening	Closure	Comment
H^+^; CO_2_		C	takes 7–10 min.
Na^+^, Ca^2+^		C	e.g., in hypoxia
acylcarnitine, lysophosphoglycerides		C	in ischemia/ reperfusion injury
ATP loss		C	in ischema
arachidonic acid, oleic acid, palmitoleic acid, hepantol, octanol, narcotics (halothane)		C	via incorporation? or via altered phosphorylation?
Unspecific PKC stimulation (e.g., phorbol esters)		C	depends on PKC isoform present in the targeted cell
cAMP and direct or indirect stimulators of adenylylcyclase (isoprenalin, forskolin)	O		cAMP activates PKA, in Cx40 or Cx45 coupled cells
PKA	O		in Cx40 or Cx45 coupled cells
antiarrhythmic peptides, AAP10, rotigaptide, danegaptide	O		via PKCα

**Table 2 cells-10-02422-t002:** Pathways regulating the expression of cardiac connexins.

Stimulus	Enhanced Number	Enhanced Degradation	Comment
phenylephrine (α-adrenergic stimulator)	Cx43		via PKC, MAPKs (cardiomyocytes)
isoprenaline (β-adrenergic stimulator)	Cx43		via PKA, MAPKs (cardiomyocytes)
angiotensin-II	Cx43		via AT_1_ receptors (cardiomyocytes)
Endothelin	Cx43		via ET_A_-receptors (cardiomyocytes)
thyroid hormone	Cx40, Cx43		mechanism unclear
VEGF	Cx43		via TGFβ
bFGF		Cx43	via PKCε
EGF		Cx43	enhanced internalization
Nicotine		Cx37, Cx43	via nACh-R (endothelium)

## Data Availability

Not applicable.

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
