# Peer review of "Remodeling of Cardiac Gap Junctional Cell–Cell Coupling"

_cells, 2021, doi:10.3390/cells10092422_

Round 1

Reviewer 1 Report

The review  “Remodeling of cardiac gap junctions by Dhein and Salammeh is a well organized review focused on remodeling/distribution of gap junctions as an important element in arrhythmias.  A number of related topics are discussed. 

There are only a few questions and concerns:

page 1 last paragraph: It would be best to define what is meant by “spread of an action potential”. It might be assumed to  mean conduction/propagation but the authors should clarify.

Within the same paragraph the authors sight Weingart and Maurer (1988) noting that only a few gap junction channels are needed for “spread”.  What should also be discussed is the relationship between functioning channel number and conduction.  

Page 2:  acute opening and closure implies non-acute or slow opening and closure.  might be better to say channel open and closed states or some equivalent. The authors have generated tables to show how specific agents either result in opening or closing— this makes it necessary to define open and closed— acute is not the best word.

the literature sited for acute processes (author misspelled) is inadequate. A number of papers have been published defining the gating process for Cx43 for example.  In the context of all the agents able to affect channel function some of the basics should be referenced/discussed. A number of authors have qualitatively determined open probabilities, mean open time and closed time and in some cases shown the effects of specific agents on open time and closed not to mention open probability, pH being a classic

Another aspect important to understanding how many channels are functioning and under what anatomical construct focuses on the number of channels at anyone interphase that are functioning.  Bukauskus et al (PNAS) 2000 paper addresses this issue which is essential to ultimately understanding cardiac AP conduction.  Hence a brief discussion would be appropriate especially in the context of the discussion on page 4.  

Two references should be added: Carmeliet E. 2019 Physiol Report. The author has written and excellent review with a concise history of AP conduction in the heart and also discusses ephaptic transmission.  

the authors mention ephaptic transmission in the text but it would be useful to discuss the concept with regards to arrhythmias. 

there is a very recent paper in J Gen Physiol entitled “Intercalated disk nanoscale structure regulates cardiac conduction” by Struckman et al.  Clearly this paper is important to understanding remodeling and should be sited. 

Author Response

We wish to thank both reviewers and the editors for their helpful comments and interesting ideas which have helped to improve the paper. We have amended all points raised by the referees and marked them in yellow in the new version. Please find below a point-to-point letter stating how we dealt with each point.

With many thanks I remain

Yours sincerely

Stefan Dhein

Reviewer 1

The review  “Remodeling of cardiac gap junctions by Dhein and Salameh is a well organized review focused on remodeling/distribution of gap junctions as an important element in arrhythmias.  A number of related topics are discussed. 

There are only a few questions and concerns:

page 1 last paragraph: It would be best to define what is meant by “spread of an action potential”. It might be assumed to  mean conduction/propagation but the authors should clarify.

=>”successful cell-to-cell conduction”

Within the same paragraph the authors sight Weingart and Maurer (1988) noting that only a few gap junction channels are needed for “spread”.  What should also be discussed is the relationship between functioning channel number and conduction.  

=> The number of gap junction channels is inversely correlated to the conduction velocity (Seidel et al., 2010). However, under normal normoxic conditions, some experiments have shown that a 90% decrease in the number of gap junctions is necessary to reduce conduction velocity by 25% (Jongsma & Wilders, 2000). This oversimplifies the situation a bit, since the effectivity of conduction not only depends on the pure number of gap junction channels but also their single channel conductance, on the relationship between this number of channels and the sizes of the activated cell and of the cell /area which is to be activated.

Page 2:  acute opening and closure implies non-acute or slow opening and closure.  might be better to say channel open and closed states or some equivalent. The authors have generated tables to show how specific agents either result in opening or closing— this makes it necessary to define open and closed— acute is not the best word.

the literature sited for acute processes (author misspelled) is inadequate. A number of papers have been published defining the gating process for Cx43 for example.  In the context of all the agents able to affect channel function some of the basics should be referenced/discussed. A number of authors have qualitatively determined open probabilities, mean open time and closed time and in some cases shown the effects of specific agents on open time and closed not to mention open probability, pH being a classic

page 3, 1st para:

=>Gap junction channels can switch between various states: a closed state, an open state with several subconductance states (e.g.in the case of Cx43 three: 30, 61, 89 pS (Kwak et al., Mol Biol Cell 1995), a closed state (0 pS), and a residual state (3 pS; Valiunas et al., 1997) (See also Bukauskas & Verselis, 2004). The preferential subconductance state seems to depend on the phosphorylation / de-phosphorylation status of the connexin at certain residues, in particular within the C-terminal. Besides the single channel conductance also the mean open time and open probability (Brink et al., 1996) and the transition time (from open to closed state and vice versa) can be regulated (Bukauskas & Peracchia, 1997). Generally, the macroscopic conductance (gj) is affected by the number of channels (N), their unitary conductance (gj) and their open probability (Po) as given by the equation: gj = N x gj x Po. Important regulators of gap junction conductance are pH, intracellular calcium and transjunctional voltage. Thus, intracellular acidification results in closure of gap junction channels. Among others, two main processes are responsible: upon drop in pH the C-terminal changes its conformation thereby closing the channel pore in a “ball-and-chain-like mechanism (Ek-Vitorin et al., 2016), and secondly H+ protons can displace Ca2+ from its binding sites leading indirectly to gap junction closure via increased [Ca2+] (Peracchia, 2004). On the other hand, the N-terminal is involved in voltage sensing and connexin channel gating (Verselis et al., 1994). Closure of Cx43 gap junction channels by elevated [Ca2+]i is thought to be mediated via Ca2+-calmodulin signaling and a calmodulin interactive site at the C-terminal (Peracchia, 2004). Phosphorylation of the C-terminal can change macroscopic gap junction channel conductance (Moreno & Lau, 2007). PKC has been reported to (a) reduce the unitary event activity from 100 pS to 50-60 pS events (Lampe et al., 2000) and (b) to increase the open probability (Kwak et al., 1995; Kwak & Jongsma, 1996), so that the resulting net effect is a mixture, which may depend on the PKC isoforms involved. For more details on this issue see the review by Leybaert et al. (Leybaert et al., 2017).

page 3, 2nd para: Misspelling is corrected =>  Leybaert

Kwak BR, Hermans MM, De Jonge HR, Lohmann SM, Jongsma HJ, Chanson M. Differential regulation of distinct types of gap junction channels by similar phosphorylating conditions. Mol Biol Cell. 1995 Dec;6(12):1707-19. doi: 10.1091/mbc.6.12.1707. PMID: 8590800; PMCID: PMC301327.

Valiunas V, Bukauskas FF, Weingart R. Conductances and selective permeability of connexin43 gap junction channels examined in neonatal rat heart cells. Circ Res. 1997 May;80(5):708-19. doi: 10.1161/01.res.80.5.708. PMID: 9130452.

Bukauskas FF, Verselis VK. Gap junction channel gating. Biochim Biophys Acta. 2004 Mar 23;1662(1-2):42-60. doi: 10.1016/j.bbamem.2004.01.008. PMID: 15033578; PMCID: PMC2813678.

Brink PR, Ramanan SV, Christ GJ. Human connexin 43 gap junction channel gating: evidence for mode shifts and/or heterogeneity. Am J Physiol. 1996 Jul;271(1 Pt 1):C321-31. doi: 10.1152/ajpcell.1996.271.1.C321. PMID: 8760061.

Bukauskas FF, Peracchia C. Two distinct gating mechanisms in gap junction channels: CO2-sensitive and voltage-sensitive. Biophys J. 1997 May;72(5):2137-42. doi: 10.1016/S0006-3495(97)78856-8. PMID: 9129815; PMCID: PMC1184407.

Ek Vitorín JF, Pontifex TK, Burt JM. Determinants of Cx43 Channel Gating and Permeation: The Amino Terminus. Biophys J. 2016 Jan 5;110(1):127-40. doi: 10.1016/j.bpj.2015.10.054. PMID: 26745416; PMCID: PMC4805864.

Peracchia C. Chemical gating of gap junction channels; roles of calcium, pH and calmodulin. Biochim Biophys Acta. 2004 Mar 23;1662(1-2):61-80. doi: 10.1016/j.bbamem.2003.10.020. PMID: 15033579.

Verselis VK, Ginter CS, Bargiello TA. Opposite voltage gating polarities of two closely related connexins. Nature. 1994 Mar 24;368(6469):348-51. doi: 10.1038/368348a0. PMID: 8127371.

Moreno AP, Lau AF. Gap junction channel gating modulated through protein phosphorylation. Prog Biophys Mol Biol. 2007 May-Jun;94(1-2):107-19. doi: 10.1016/j.pbiomolbio.2007.03.004. Epub 2007 Mar 15. PMID: 17507079; PMCID: PMC1973155.

Lampe PD, TenBroek EM, Burt JM, Kurata WE, Johnson RG, Lau AF. Phosphorylation of connexin43 on serine368 by protein kinase C regulates gap junctional communication. J Cell Biol. 2000 Jun 26;149(7):1503-12. doi: 10.1083/jcb.149.7.1503. PMID: 10871288; PMCID: PMC2175134.

Kwak BR, van Veen TA, Analbers LJ, Jongsma HJ. TPA increases conductance but decreases permeability in neonatal rat cardiomyocyte gap junction channels. Exp Cell Res. 1995 Oct;220(2):456-63. doi: 10.1006/excr.1995.1337. PMID: 7556455.

Kwak BR, Jongsma HJ. Regulation of cardiac gap junction channel permeability and conductance by several phosphorylating conditions. Mol Cell Biochem. 1996 Apr 12-26;157(1-2):93-9. doi: 10.1007/BF00227885. PMID: 8739233.

Another aspect important to understanding how many channels are functioning and under what anatomical construct focuses on the number of channels at anyone interphase that are functioning.  Bukauskus et al (PNAS) 2000 paper addresses this issue which is essential to ultimately understanding cardiac AP conduction.  Hence a brief discussion would be appropriate especially in the context of the discussion on page 4.

Page 5, 2nd para:

=> Using an EGFP-tagged Cx43 (EGFP = enhanced green fluorescent protein) Bukauskas and colleagues (2000) showed that channels from plaques by clustering and that in plaques with about 0.5 µm diameter about 2000 channels can be found of which 10 to 20% are functional. In smaller plaques this fraction is lower and tends to zero. These authors assumed that channels may cycle between an active and inactive state, which may be regulated by phosphorylation processes. Critically, one needs to state, that it cannot be totally excluded that the EGFP-tag itself may affect connexin life cycle, availability of phosphorylation sites or geometrical features regarding the clustering in plaques. However, these observations suggest that very small plaques of connexins might be too small to contain a sufficient number of function channels for allowing action potential transfer.

Bukauskas FF, Jordan K, Bukauskiene A, Bennett MV, Lampe PD, Laird DW, Verselis VK. Clustering of connexin 43-enhanced green fluorescent protein gap junction channels and functional coupling in living cells. Proc Natl Acad Sci U S A. 2000 Mar 14;97(6):2556-61. doi: 10.1073/pnas.050588497. PMID: 10706639; PMCID: PMC15967.

Two references should be added: Carmeliet E. 2019 Physiol Report. The author has written and excellent review with a concise history of AP conduction in the heart and also discusses ephaptic transmission.  

the authors mention ephaptic transmission in the text but it would be useful to discuss the concept with regards to arrhythmias. 

there is a very recent paper in J Gen Physiol entitled “Intercalated disk nanoscale structure regulates cardiac conduction” by Struckman et al.  Clearly this paper is important to understanding remodeling and should be sited. (=> Moise et al., 2021; see below)

=>Both references have been included

End of page 1 / 1st para, p.2:

Recent studies showed that ephaptic coupling requires an electrical field sufficient for activation of clusters of sodium channels, and that the clustering of sodium channels potentiates the ephaptic interaction, which is even higher if the gap junction coupling is reduced (Hirchi et al. 2018). This may contribute to the formation of areas with slow conduction as a typical arrhythmogenic factor (Carmeliet, 2019).

Hichri E, Abriel H, Kucera JP. Distribution of cardiac sodium channels in clusters potentiates ephaptic interactions in the intercalated disc. J Physiol. 2018 Feb 15;596(4):563-589. doi: 10.1113/JP275351. Epub 2018 Jan 9. PMID: 29210458; PMCID: PMC5813604.

Carmeliet E. Conduction in cardiac tissue. Historical reflections. Physiol Rep. 2019 Jan;7(1):e13860. doi: 10.14814/phy2.13860. PMID: 30604919; PMCID: PMC6316167.

Page 5, 3rd para:

Gap junction channels are clustered in the intercalated disk (ID) in close neighborhood to sodium channels (Moise et al., 2021). This would allow the elicitation of an action potential at the earliest moment when the current flowing through the gap junctions depolarizes the membrane in the ID area. However, the ID is not a simple homogenous structure but shows a complicated pattern with nanodomains which can enhance or attenuate gap junction coupling. Moreover, this explains that a successful transfer of action potentials is not simply granted by a certain number of gap junctions, but the interaction of ID nanodomains, gap junction clustering and interaction with neighbored sodium channels (Moise et al., 2021).

Moise N, Struckman HL, Dagher C, Veeraraghavan R, Weinberg SH. Intercalated disk nanoscale structure regulates cardiac conduction. J Gen Physiol. 2021 Aug 2;153(8):e202112897. doi: 10.1085/jgp.202112897. Epub 2021 Jul 15. PMID: 34264306; PMCID: PMC8287520.

Reviewer 2 Report

Authors Dhein and Salameh cover the changes occurring in the cardiac conduction system and associated cardiomyocytes during disease condition. Specifically, the authors focus on the gap junction proteins (Connexins) and the role that these proteins play in transmitting electrical signals through the heart.

Overall, the paper is well written and the level of depth of discussion is appropriate. The two figures and two tables are useful and helpful for readers. Authors point out shortcomings in knowledge in the field and suggest future steps to address these.

Some minor sentence structure and run-on sentences that can be fixed with editing for language. 

Author Response

We wish to thank both reviewers and the editors for their helpful comments and interesting ideas which have helped to improve the paper. We have amended all points raised by the referees and marked them in yellow in the new version. Please find below a point-to-point letter stating how we dealt with each point.

With many thanks I remain

Yours sincerely

Stefan Dhein

Reviewer 2

Authors Dhein and Salameh cover the changes occurring in the cardiac conduction system and associated cardiomyocytes during disease condition. Specifically, the authors focus on the gap junction proteins (Connexins) and the role that these proteins play in transmitting electrical signals through the heart.

Overall, the paper is well written and the level of depth of discussion is appropriate. The two figures and two tables are useful and helpful for readers. Authors point out shortcomings in knowledge in the field and suggest future steps to address these.

Some minor sentence structure and run-on sentences that can be fixed with editing for language. 

=>The paper has been seen by an English colleague.

Round 2

Reviewer 1 Report

The revision is very well done.  It represents and excellent addition to our understanding of the intercalated disc and gap junctions